# Recognition of Rare Microfossils Using Transfer Learning and Deep Residual Networks

**DOI:** 10.3390/biology12010016

**Published:** 2022-12-21

**Authors:** Bin Wang, Ruyue Sun, Xiaoguang Yang, Ben Niu, Tao Zhang, Yuandi Zhao, Yuanhui Zhang, Yiheng Zhang, Jian Han

**Affiliations:** 1School of Information Science & Technology, Northwest University, Xi’an 710069, China; 2Shaanxi Key Laboratory of Early Life and Environments, State Key Laboratory of Continental Dynamics, Department of Geology, Northwest University, Xi’an 710069, China; 3State Key Laboratory of Palaeobiology and Stratigraphy (Nanjing Institute of Geology and Palaeontology, CAS), Nanjing 210008, China

**Keywords:** early Cambrian, microfossils, small sample, transfer learning, residual network

## Abstract

**Simple Summary:**

The collection of early Cambrian microfossils leads to the amassing of a pile of thousands of tiny tubes, grains and various fragments. Rare type of microfossils with high academic value are mingled with numerous ordinary fossils and the traditional way of manual selection is very inefficient. Many artificial intelligence (AI) technologies have already been applied in fossil image recognition, but current methods largely depend on a great number of fossil images to “train” the AI model. However, usually only a handful of samples are available for specific rare fossil taxa and these cannot provide enough photos for AI. In this study, we fine-tuned a new convolutional neural network, combining pre-trained models from a nature image database to solve the problem of the lack of training materials. Through many tests, this new model was proved valid. It presented relatively high accuracies in recognizing specific micro fossil taxa, while the required number of corresponding fossil images is very low.

**Abstract:**

Various microfossils from the early Cambrian provide crucial clues for understanding the Cambrian explosion and the origin of animal phyla. However, specimens with important anatomical structures are extremely rare and the efficiency of retrieving such fossils by traditional manual selection under a microscope is quite low. Such a contradiction has hindered breakthroughs in micropaleontology for a long time. Here, we propose a solution for identifying specific taxa of Cambrian microfossils using only a few available specimens by transferring a model pre-trained on natural image datasets to the field of paleontological artificial intelligence. The method employs a 34-layer deep residual neural network as the underlying framework, migrates the ImageNet pre-trained model, freezes the low-layer network parameters and retrains the high-layer parameters to build a microfossil image recognition model. We built training sets with randomly selected images of varied number for each taxon. Our experiments show that the average recognition accuracy for specific taxa of Cambrian microfossils (50 images for each taxon) is higher than 0.97 and it can reach 0.85 with only three training samples per taxon. Comparative analyses indicate that our results are much better than those of various prevalent methods, such as the transpose convolutional neural network (TCNN). This demonstrates the feasibility of using natural images (ImageNet) for the training of microfossil recognition models and provides a promising tool for the discovery of rare fossils.

## 1. Introduction

The world-wide sudden appearance of microfossils near the Ediacaran-Cambrian boundary presents uncontroversial records of the first major phyla-level diversification of metazoans, that is, cnidarians, scalidophorans, lophotrochozoans and many problematic forms [1]. Extending the fossil records of all animal phyla to the Ediacaran-Cambrian boundary or deeper episodes has been a long-running task in paleontological research. The Kuanchuanpu Biota (Cambrian Fortunian, ca. 535 Ma) in southern Shaanxi, China has attracted worldwide attention because of its exceptional three-dimensional preservation of a bulk of secondarily phosphatized microfossils, including cyanobacteria [2], multicellular algae [3], possible protozoans [4], as well as some unmineralized metazoans, such as cnidarians [5,6], scalidophorans [7] and primitive bilaterians [8,9]. Most of these metazoan fossils represent the earliest fossil records of animals; thus, the Kuanchuanpu Biota provides a critical window into the early Cambrian benthic marine ecosystem. Similar to the research history of other fossil lagerstätten, those taxa, having plentiful fossil materials, have been investigated extensively for decades, while the fossil taxa deficient in samples, which usually provide key evidence for understanding animal evolution [10,11], are very difficult to address in terms of anatomy and affinity. However, in the process of manual specimen selection under microscopes, the finding of more microscopic specimens of rare taxa is severely obstructed by countless dominant species, that is, biomineralized tubular fossils. Therefore, it is highly desirable to introduce artificial intelligence technology to solve the problem of finding rare but important microfossils.

Some basic machine learning methods have already been applied in the identification of fossils of large quantity, such as conodonts and pollen fossils [12,13]. Our team has also successfully combined the histogram of oriented gradient (HOG), watershed algorithm and scale-invariant feature transform (SIFT) with support vector machine (SVM) classifiers to identify early Cambrian microfossils [14]. Convolutional neural networks (CNNs), which simulate the analysis and learning ability of the human brain by constructing a complex neural network structure and increase accuracy and efficiency as a consequence [15,16,17,18], have become mainstream methods for fossil identification [19,20,21,22,23,24].

The problem of traditional fossil identification models that use CNNs is that they are extremely dependent on a large number of fossil specimens, namely the training data. Using only a few specimens leads to poor performance or even dysfunction. Apparently, this drawback does not serve the purpose of our project, which is to find new and rare fossil types automatically. Therefore, we attempt to resolve this conflict by combining a deep residual network (ResNet) with transfer learning. ResNet is a derived type of CNN that optimizes the fitting of deep models and reduces the training time and error as the network depth increases [25]. Transfer learning [26,27] can compensate for the shortage of rare fossil data by enabling the migration of pre-trained models based on a readily available dataset, such as ImageNet [28], which is currently the world’s largest natural image recognition database and includes more than 14 million images and 20,000 categories (http://www.image-net.org, accessed on 16 December 2021). By fine-tuning the network parameters of the ImageNet pre-trained model using the microfossil datasets, we tested the recognition performance for nine taxa in the Kuanchuanpu Biota with very limited training specimens (Figure 1, Table 1). The model showed promising feasibility for rare fossil identification.

## 2. Materials and Methods

### 2.1. Hardware and Software Environment

In this study, the training and test codes were based on Python 3.7 (Creator: Python Software Foundation, Location: Beaverton, OR, USA) and PyTorch 1.7 (Creator: Facebook, Inc., Location: Menlo Park, CA, USA), with the initial learning rate set to 0.0001 and the batch size set to 16. We used the Adam algorithm to optimize the network parameters and ran it on a Dell T7600 graphics workstation with a Windows 10 Education 64-bit OS, two Intel Xeon Gold 5118 processors, 128 GB RAM and an 11 GB NVIDIA GeForce RTX 2080Ti GPU. The corresponding versions of NVIDIA CUDA (Creator: NVIDIA Corporation, Location: Santa Clara, CA, USA) and cuDNN (Creator: NVIDIA Corporation, Location: Santa Clara, CA, USA) are 10.0 and 7.4, respectively.

### 2.2. Data Acquisition

The rock samples came from Bed 2 of the Cambrian Kuanchuanpu Formation at the Hexi section of Zhangjiagou, Dahe Town, Xixiang County, southern Shaanxi Province [14]. The phosphatized microfossils were liberated from the phosphatic limestone using 10% acetic acid digestion and they were manually checked under a microscope. Most of the insoluble residues are rock fragments and a small portion are microfossils with a size range of 0.1–20 mm. The microfossil specimens were white, black, or grayish with a greasy luster. As the vast majority of microfossils have suffered transportation before bury, they thus appeared as more or less incomplete fragments.

Three-dimensionally preserved microfossils were photographed under a Leica M205C stereo microscope (Leica Microsystems, Singapore) at a scale of 1 mm or 500 μm using a low magnification (×10) method to ensure depth of field. One microfossil specimen yielded one image and all images were captured with the same blue background to prevent background interference. Each picture is 1024 × 1024 pixels per inch (PPI) in resolution and in RGB format. We collected images of nine common taxa of microfossils [29] including *Conotheca*, *Archaeooides*, *Quadrapyrgites*, *Protohertzina*, *Maikhanella*, *Carinachites*, *Qinscolex* [30], *Hyolithellus* and irregular-shaped dross (here, treated as a taxon for convenience) (Figure 1 and Table 1).

### 2.3. Data Preprocessing

Because the number of images for various taxa varies greatly, we sampled the same number of images for each taxon to keep the data balanced. Before inputting the microfossil images into the ResNet model, we used a series of data augmentation methods to preprocess the original images, such as background removal, affine transformation (translation, rotation, shearing and scaling) and adding Gaussian noise, to increase the data volume and diversity of the training dataset and to prevent overfitting to the greatest extent (Figure 2). To prevent the effect of exposure, brightness and other factors on the image pixel values, we normalized the images after data augmentation, that is, we first derived the mean $X_{mean}$ and standard deviation $X_{std}$ of the pixel matrix of the original image X and then used $X_{\;standard}=\left(X-X_{\;mean}\right)/X_{\;std}$ to obtain the pixel matrix of the normalized image $X_{\;standard}$ to obtain the pixel matrix of each microfossil image mapped to the same scale range [31].

## 3. Network Structure

The rare microfossil recognition model was transferred from the ResNet pre-trained on ImageNet and then the network parameters were fine-tuned on our microfossil dataset (Figure 3). We selected as the backbone the ResNet with 34 layers (Figure 4), hereafter referred to as ResNet-34, which contained many similar residual units, and what they had in common was the addition of short-circuit connections (solid and dashed lines in Figure 4) in two consecutive convolutional layers. These units effectively solve the problem of gradient disappearance or gradient explosion during the backpropagation of updated parameters. ResNet-34 has four residual blocks, namely Conv2_x, Conv3_x, Conv4_x and Conv5_x [25,32].

In this study, we pre-trained the ResNet-34 model on 1000 image classes (approximately 1000 images per class) in ImageNet [28,33] and then constructed a microfossil recognition model by parameter transfer, that is, by first retaining the network structure and parameters of some layers and then adding a small amount of microfossil data and retraining the network structure of the remaining layers to fine-tune the parameters [34,35]. Here, we employed three strategies for fine-tuning the parameters of the ResNet-34 model, as shown in Table 2.

TF-ResNet (ResNet + Train Full Connection Layers): Only the full connection layers are trained and the parameters of the remaining layers of the pre-trained model are retained.TS-ResNet (ResNet + Train Specific Characteristic Layers): The feature extraction layer of the pre-trained model is divided into two parts: general feature layers (learning edge, texture and color features) and special feature layers (learning more abstract internal features), keeping the parameters of the bottom general feature layers and retraining the parameters of the top special feature layers.TA-ResNet (ResNet + Train All Layers): A pre-trained model structure is used and all the entire network parameters are retrained based on the microfossil dataset.

## 4. Experiments and Results

We randomly selected 100 images of each taxon from the microfossil microscopic images (Table 1) collected and processed by data augmentation in the previous stage to form a unique test set. For training sets, we randomly selected a specific number of images (1, 3, 5, 10, 50, 100, 200) from each class as one group and a total of 10 groups of train sets were built to test the repeatability. To make the experimental conclusions convincing, we used two metrics commonly used in confusion matrices, precision and recall, to evaluate the precision and coverage of the model prediction results [22,36] and the average scores of multiple experiments based on 10 groups of training sets were chosen as the final result. Precision indicates the ratio of the number of fossils of a certain taxa correctly predicted to the number of fossils predicted to be of that taxon, and recall indicates the ratio of the number of fossils of a certain taxa correctly predicted to the actual number of fossils of that taxon. Usually, precision and recall are correlated; the higher the precision, the lower the recall and vice versa. The aim of this study was to find more fossils of the same taxa as accurately as possible based on a small number of microfossil specimens, which means that precision and recall are equally important. Therefore, the harmonized average of the two (F1-score) was introduced to further evaluate the combined performance of the model. The formulae for the three evaluation metrics are presented in Table 3.

### 4.1. Evaluation of Transfer Strategies

We selected 200 images from each of the nine taxa of microfossil images as the training set and compared the differences in performance between the initial ResNet model and the three ResNet models with different transfer strategies (TF-ResNet, TS-ResNet and TA-ResNet in Table 2) using a unique test set (Figure 5 and Appendix A). Only the optimal solution of TS-ResNet is shown in Appendix A, where Conv2_x is retained as a generic feature layer with parameters and Conv3_x to Conv5_x is retrained with parameters as specific feature layers. The test results for the remaining solutions are presented in Appendix A.

Our experiments indicate that the application of transfer strategies can effectively improve the performance of neural networks and that the average F1-score of TF-ResNet, TS-ResNet and TA-ResNet were all better than those of ResNet (Appendix A). Among the three transfer strategies, TS-ResNet exhibits the best F1-score (0.959–1.00, average 0.98) for each taxon, while TF-ResNet and TA-ResNet show evident fluctuations depending on the taxon of the microfossils and the recognition accuracy of TF-ResNet is relatively low (Figure 5). In summary, TS-ResNet demonstrates almost perfect accuracy and stability in recognizing the classification tasks covering multiple taxa of microfossils. Therefore, in subsequent experiments, we chose TS-ResNet as the parameter fine-tuning strategy.

### 4.2. TS-ResNet Performance on Training Datasets of Varied Sizes

In the TS-ResNet model, we built different training datasets with 1, 3, 5, 10, 50, 100 and 200 images of each taxon and tested the impact of training dataset sizes on the recognition performance (Figure 6, Appendix A). When the training datasets per taxon exceeded 50 images, TS-ResNet exhibited a misjudgment rate of less than 0.03. Nearly all testing images were correctly predicted and classified and the F1-score of various taxa fluctuated slightly (0.975 ± 0.025). As the training dataset sizes decreased to three images per taxon, the accuracies for most taxa remained at a high level (>0.90). Only three taxa (*Qinscolex*, *Hyolithellus* and dross) showed a reduction in recognition performance, but the accuracy was still greater than 0.648. In the case of one image per taxon, the F1-score of all taxa dropped clearly (average F1-score 0.66).

### 4.3. Performance Comparison of TS-ResNet with Other Models

To verify the performance advantages of TS-ResNet in identifying multi-taxon microfossils under small-sample conditions, we used several methods of machine learning and deep learning to perform ablation experiments with TS-ResNet. Machine learning is a method that combines HOG feature extraction and an SVM classifier to identify single microfossils [14]. Deep learning is widely used with VGG and InceptionV3, which are suitable for the introduction of transfer learning [37]. Based on the VGG and InceptionV3 models pre-trained on ImageNet, we adopted the same transfer strategy, parameter settings, training set and testing set as for TS-ResNet in order to obtain the corresponding TS-VGG and TS-InceptionV3 and we recorded the resulting values under the optimal layer structure division.

The average F1-scores of all three deep neural networks with transfer strategies were better than those of the machine learning method (HOG + SVM) using the same training dataset size. In the case of few training images (1 or 3), the machine learning method is invalid and the performance is still poor (0.56) when the training dataset increases to 10 images. Only with a large number of training images (at least 200) is the recognition accuracy reliable (0.927), which is consistent with previous research [14]. Other deep learning methods (TS-VGG and TS-InceptionV3), which are integrated with transfer learning, are still functional in the scenario of very limited training specimens/images (1 or 3) and require fewer training images (~10) to achieve acceptable recognition accuracies (>0.797) (Figure 7, Appendix A). When the training dataset was expanded more than 200 images, the models could recognize microfossils very accurately (TS-VGG: 0.969, TS-InceptionV3: 0.949). TS-ResNet demonstrated the highest score in all cases with different numbers of training images.

## 5. Discussion

### 5.1. Paleontological Significance of Rare Taxa Recognition

In comparison with well-documented common fossils, rare fossils in paleontology are of crucial research significance as they can fill the gaps regarding prehistoric biological affinities, paleo-ecological environments, life modes and the evolution process of earth life. Microfossils at the Ediacaran-Cambrian boundary are especially important in this case and the significance of finding more rare microfossils is as follows. (1) Classification: microfossils commonly found in the Kuanchuanpu Biota in southern Shaanxi are generally dominated by mineralized skeletons, such as the *Anabarites* and *Conotheca*. Because of the lack of soft tissue, these fossils have long been controversial in affinity [38,39], thus, to find exceptional specimens having soft body structures it is necessary to address the classification position of these fossils. (2) To trace the evolution of the Ediacaran relicts in the Cambrian: the latest Ediacaran Gaojiashan Biota in southern Shaanxi is represented by various tubular fossils, such as *Cloudina* [40]. Fortunately, we found a single specimen of *Feiyanella* from the Kuanchuanpu Biota that is highly comparable to *Cloudina* [41], therefore, the identification of more specimens of *Feiyanella* is helpful in understanding the fate and evolution of these problematical animals. (3) To find more primitive, soft-bodied forms: according to the estimation of molecular clocks, the origin of major animal phyla significantly predates fossil records of various small shell fossils from the late Ediacaran to the early Cambrian period [42,43]. Most of the Cambrian microfossils have independently acquired skeletons with different degrees of biomineralization. Their older ancestors, which lack organic or inorganic skeletons, are generally difficult to preserve as fossils [44]. These ancient forms, if occasionally recorded as microfossils, are very scarce. (4) To find the intermediate taxa between animal phyla: due to the extinction of the intermediate taxa, various living animal phyla are discernible from each other with regard to their prominent morphological gaps [1,45], which makes it difficult to judge the phylogenetic relationships among different animal phyla. These intermediate taxa, in general but not always, are often rare because of their relatively low adaptability in comparison with those of the surviving forms.

In summary, rare taxa exhibit increasing difficulty and low potential to be find by manual identification under a microscope. Therefore, the introduction of new artificial intelligence methods to solve the problem of rareness is of great significance to the study of the origin and evolution of animals.

### 5.2. Necessity and Feasibility of Transferring Pre-Trained Models Based on Natural Images

Theoretically, the performance of AI microfossil recognition models, either machine learning or deep learning, is closely related to the size of the dataset. Usually, the greater the amount of training data, the better the performance. Our experiments show that the machine learning methods based on artificially designed feature extraction patterns exhibited inferior performance in dealing with small dataset scenarios. Such patterns require humans to designate specific features from the training images. If the training dataset is small, it cannot extract a sufficient number of features to cover the possible similarities in the test dataset, thus leading to missing or misrecognition of microfossils in practice. Deep learning automatically extracts and learns various features from the training dataset through networks of different depths, improving the drawbacks of human-designed feature extraction patterns. However, a training dataset that is not large enough will also cause overfitting, gradient explosion and result in the network’s inability or failure to converge [22]. Therefore, the current deep learning models cannot be directly applied to the excavation of rare fossil taxa.

One solution for expanding training datasets having an insufficient number of images is data augmentation, as exemplified by the affine transformation, which can increase the number of images in the dataset. However, the affine transformation only generates equivalent image transformation data and does not effectively extend the feature space when the original dataset is small. In practice, the number of rare fossil taxa is usually quite small, that is, less than five specimens, so that data augmentation can only slightly alleviate the problem of overfitting.

Recently in information science, transfer learning has provided a promising solution for overcoming the scarcity of training datasets by enabling the development of a new model by migrating the parameters of models pre-trained on a readily available dataset [34,35]. In this study, the most ideal pre-trained model would be built based on corresponding microfossil images. However, the existing open-access paleontological databases GBDB (Geobiodiversity Database) (http://www.geobiodiversity.com, accessed on 9 October 2021) and PBDB (Paleobiology Database) (https://paleobiodb.org, accessed on 9 October 2021) do not contain a sufficient number of fossil images for AI training, especially for early Cambrian microfossils and they do not make available any pre-trained models for transfer learning because such tasks are not in their to-do lists. Building a new pre-trained model by ourselves based on these two databases would be time-consuming. Furthermore, the superiority of a new pre-trained model is doubtful. In contrast, in the natural image domain, there are a large number of public datasets with rich and diverse content and more options for available pre-trained models. In particular, the ImageNet dataset is popular among information science researchers and it shows great potential for geological studies [22]. The images in it are obtained under natural light, similar to our microfossil images, so that they share similar generic image features, such as spectral range. Therefore, it is relatively easy and effective to migrate a model pre-trained on the natural images of ImageNet to the Cambrian microfossil artificial intelligence recognition task. Our experimental results demonstrate that, with a small number of microfossil training images, this transfer approach significantly improves the recognition performance of deep neural networks and achieves cross-domain (natural images vs. microfossil images) transfer learning, which provides a new and reliable solution for the future practice of rare fossil discovery.

### 5.3. Performance Analysis of TS-ResNet

As demonstrated in this study, the models adopting transfer learning (TS/TF/TA-ResNet) exhibited evident advantages compared with ResNet. Even under the circumstances of relatively plentiful training datasets (200 microfossil images per taxon), transfer learning networks performed better than ResNet, especially for taxa with varying morphologies (*Hyolithellus* and dross) (Figure 5). One possible underlying reason is that ResNet without transfer learning prevented the learning of features from all the sample morphologies of some taxa (i.e., *Hyolithellus* and dross), which resulted in a prominent difference in precision and recall and hampered the overall performance.

By using transfer learning, therefore, we preserved some of the generic feature parameters learned on natural images and only used a small number of microfossil images to retrain the high-layer network and fine-tune specific feature parameters, and consequently improved the recognition accuracy of the models for those microfossils (*Hyolithellus* and dross) having large morphological variations. The average identification accuracies of all three transfer models (TS/TF/TA-ResNet) for the nine microfossil taxa were higher than 0.90 (Figure 5 and Appendix A). Among three transfer models, the transfer strategy of TS-ResNet achieved optimal performance by dividing the appropriate generic and specific feature layers, whereas TF-ResNet only fine-tuned the parameters of the fully connected layer, which accounted for a very small proportion relative to the whole model. TA-ResNet used a few of microfossils to adjust all model parameters insufficiently. In summary, both TF-ResNet and TA-ResNet exhibited relatively inferior recognition performances compared with TS-ResNet.

Current baseline methods (i.e., transposed convolutional neural networks (TCNN)) still require a relatively large training dataset, for example, at least 100 images [21]. Transfer learning strategies have been introduced to reinforce datasets with thousands of microfossil images [22], but the ultimate ability of transfer learning to reduce the size of the training set has not yet been tested. In our experiment, the combination of ResNet and transfer learning (TS-ResNet) exhibited a competitive recognition capability for the task of multi-taxon microfossils. When the number of training images for each taxon decreased to as little as three, the average recognition rate remained greater than 0.85 and reached 0.97 as the number of images increased to 50. In the extreme case of a single image for each taxon, the recognition performance of TS-ResNet fluctuated; notably, the recognition rates of half of the taxa were over 0.70 (Figure 6, Appendix A).

Under small sample conditions (≤10 images per taxon), the recognition performance of TS-ResNet was closely related to the biological and taphonomical attributes of fossils, especially completeness, flexibility and morphological stability, and complexity. For taxa with rigid exoskeletons (*Protohertzina*, *Maikhanella*, *Carinachites*) and, accordingly, more intact preservation, TS-ResNet ensured higher recognition accuracies. In contrast, *Qinscolex*, which has a flexible integument [46], is usually preserved as deformed fragments. Owing to its poor completeness and strong flexibility, its recognition accuracy is lower than that of the rigid forms. The models worked perfectly when dealing with fossils having simple morphologies and little geometric variation (*Conotheca*, *Archaeooides*), while the irregular-shaped dross caused an evident drop in performance (Figure 6, Appendix A). However, all the studied fossils were retrieved from the same locality with uniform preservation condition, so they could be free of some major taphonomical differentiations, three-dimensional or flattened, phosphatized, carbonaceous or siliceous [47], which would substantially interfere with AI recognition.

## 6. Conclusions

In this study, a purposefully tuned ResNet architecture implemented using transfer learning demonstrated stronger ability to extract features of microfossil images and showed lower dependence on the size of the training dataset than did current prevalent methods. The average recognition accuracies for nine taxa were over 0.97 when the models were run on small datasets (50 images per fossil taxon) and reached 0.85 when the training sets were reduced to three images per fossil taxon. This capacity is crucial for overcoming the scarcity of those problematical microfossil taxa. This study laid the foundation for rapid automatic microfossil selection, which would liberate paleontologists from labor-intensive work and subsequently stimulate research on the origin and evolution of animal phyla by making the collection of rare and high-value fossil materials more efficient. Furthermore, in comparison with other machine learning and deep learning methods, the recognition model proposed here is also more robust and has an improved generalization ability. It exhibits great potential for applications with other geological data. In addition, such an approach could be widely introduced to any other scientific research and practical operation concerning with “small sample” problem.

## Figures and Tables

**Figure 1 biology-12-00016-f001:**
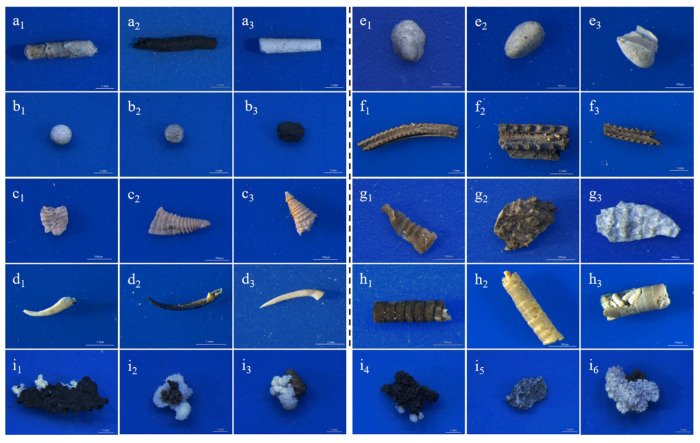
Image information on nine microfossil taxa. (**a1**–**a3**) *Conotheca* (scale bar: 1 mm); (**b1**–**b3**) *Archaeooides* (scale bar: 1 mm); (**c1**–**c3**) *Quadrapyrgites* (scale bar: 500 μm); (**d1**–**d3**) *Protohertzina* (scale bar: 1 mm); (**e1**–**e3**) Maikhanella (scale bar: 500 μm); (**f1**–**f3**) *Carinachites* (scale bar: 1 mm); (**g1**–**g3**) *Qinscolex* (scale bar: 500 μm); (**h1**–**h3**) *Hyolithellus* (scale bar: 500 μm); (**i1**–**i6**) Irregular-shaped dross (scale bar: 1 mm).

**Figure 2 biology-12-00016-f002:**
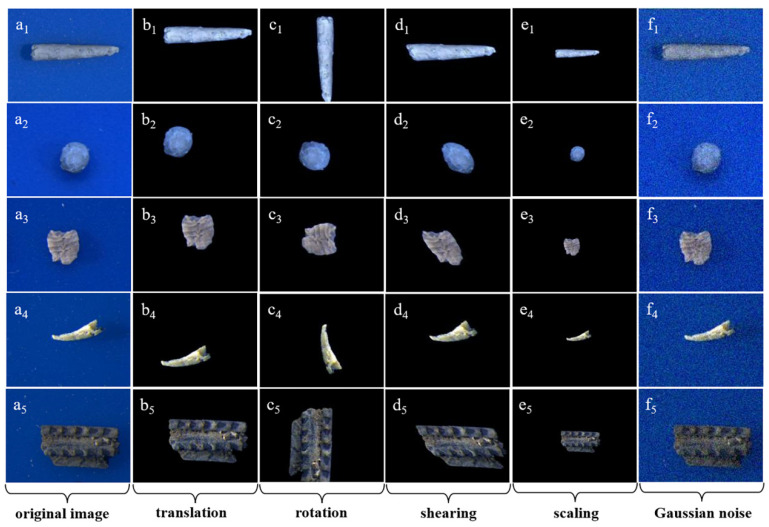
Images generated by data augmentation of some microfossils. (**a1**–**a5**) original images; (**b**–**e**) images after affine transformation (**b1**–**b5**: translation, **c1**–**c5**: rotation, **d1**–**d5**: shearing, **e1**–**e5**: scaling) based on removing background; (**f1**–**f5**) images after adding Gaussian noise.

**Figure 3 biology-12-00016-f003:**
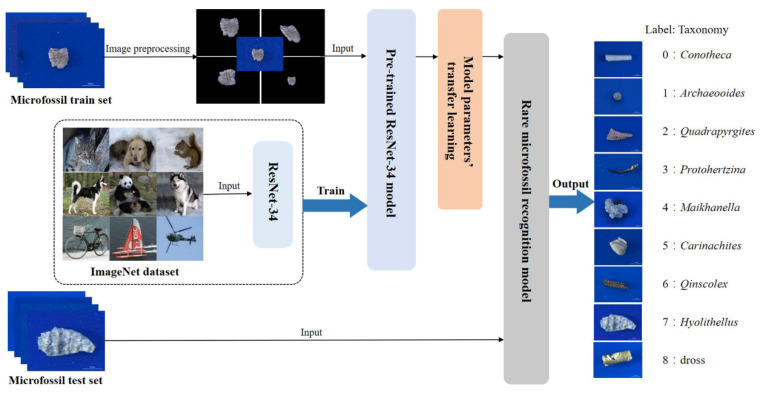
Rare microfossil recognition model based on transfer learning and ResNet-34.

**Figure 4 biology-12-00016-f004:**
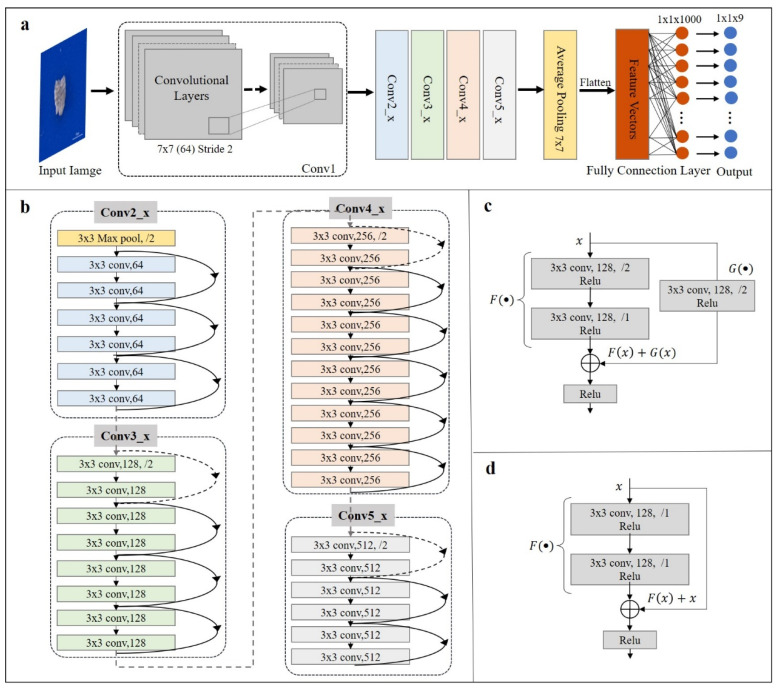
Schematic of ResNet-34. (**a**) The overall structure of ResNet-34. It consists of a convolutional unit (Conv1) and four residual blocks (Conv2_x, Conv3_x, Conv4_x and Conv5_x), which generate the final probability vectors and output the category labels by average pooling layer and fully connected layer; (**b**) Internal structure of the four residual blocks. Dashed and solid lines between layers represented two different short-circuit connection mechanisms of the residual units; (**c**) The short-circuit connection mechanism (taking Conv3_x block as an example) indicates that the number of input and output channels are different, corresponding with dashed lines in (**b**); (**d**) The short-circuit connection mechanism (taking Conv3_x block as an example) indicates that the number of input and output channels are the same, corresponding with solid lines in (**b**).

**Figure 5 biology-12-00016-f005:**
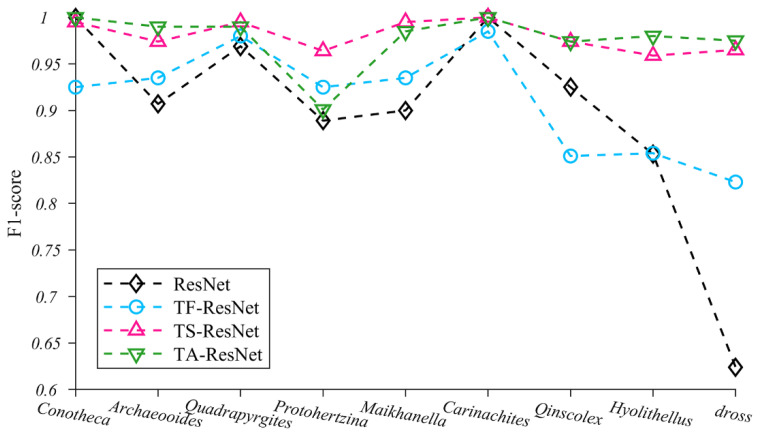
F1-score changes of nine microfossils using four training methods.

**Figure 6 biology-12-00016-f006:**
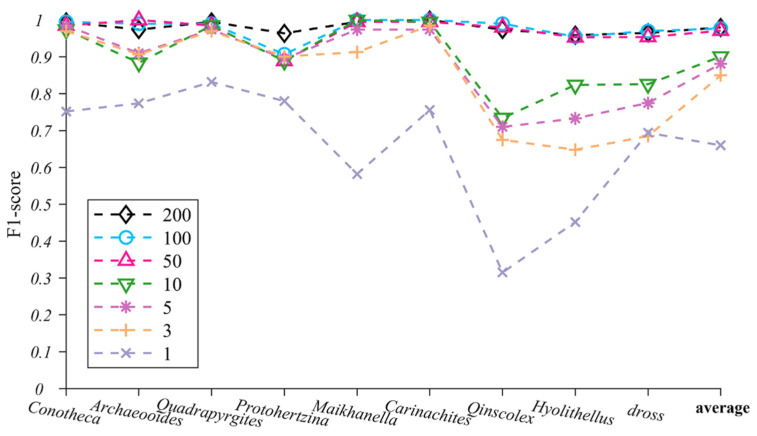
TS-ResNet’s F1-score changes for nine microfossils under different numbers of training images (the last item on the horizontal coordinate represents the average F1-score of nine microfossils).

**Figure 7 biology-12-00016-f007:**
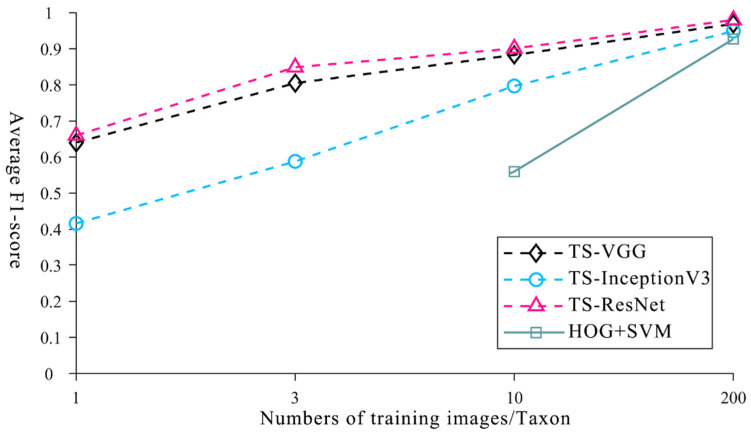
Average F1-score of machine learning, deep learning methods and TS-ResNet with different numbers of training images.

**Table 1 biology-12-00016-t001:** Basic information on nine microfossils.

Taxonomy	Shape	Size	Total Amounts
? Cnidaria: *Conotheca*	cone-shaped tube	3 mm (±2 mm)	858
Animals: *Archaeooides*	sphere	1.5 mm (±1 mm)	968
Cnidaria: *Quadrapyrgites*	cone-shaped tower	1 mm (±0.75 mm)	357
Chaetognatha: *Protohertzina*	curved spines	1.5 mm (±1 mm)	905
Mollusca: *Maikhanella*	shell	0.6 mm (±0.2 mm)	210
Cnidaria: *Carinachites*	tube with four slots	3 mm (±2 mm)	138
Scalidophora: *Qinscolex*	worms	1 mm (±0.5 mm)	56
? Annelida: *Hyolithellus*	straight, segmented tube	1.5 mm (±1 mm)	83
*null*	Irregular-shaped dross	3 mm (±2 mm)	932

Notes: ? indicates that such fossils’ biological affinities remain uncertain.

**Table 2 biology-12-00016-t002:** Three strategies for fine-tuning the parameters of the ResNet-34 pre-trained model.

Strategies	Conv1	Conv2_x	Conv3_x	Conv4_x	Conv5_x	Fc (Fully Connected Layer)
TF-ResNet	✕	✕	✕	✕	✕	✓
TS-ResNet	✕	✕	✕	✕	✓	✓
	✕	✕	✕	✓	✓	✓
	✕	✕	✓	✓	✓	✓
	✕	✓	✓	✓	✓	✓
TA-ResNet	✓	✓	✓	✓	✓	✓

Notes: ✕ indicates that the network structure and parameters of the layer were retained while ✓ indicates they were retrained.

**Table 3 biology-12-00016-t003:** Confusion matrix and evaluation index calculated accordingly.

Confusion Matrix		True Labels		Evaluation
		Positive	Negative	
Predicted	Positive	True Positive	False Positive	$P\mathrm{recision}=\frac{TP}{TP+FP}$
	Negative	False Negative	True Negative	$R\mathrm{ecall}=\frac{TP}{TP+FN}$
Evaluation		$F1-\mathrm{score}=2*\frac{P*R}{P+R}$		

## Data Availability

The data presented in this study are openly available in https://gitee.com/sw_yy/TlResNet, accessed on 13 February 2022.

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
