# Peer review of "Recognition of Rare Microfossils Using Transfer Learning and Deep Residual Networks"

_biology, 2022, doi:10.3390/biology12010016_

Round 1

Reviewer 1 Report

This is a very interesting manuscript using transfer learning to retrain some classic deep convolution neural networks to identify rare microfossils. Indeed, the transfer Learning of other classic deep learning models has advantages on small data and time saving. In this research, authors add much evidence to a field that is only going to grow in the near future. It is a valuable crossing research between deep learning and fossil records. So, I strongly suggest its publication in Biology after minor revision.

General Comments:

1.1 Title

I am not a native English speaker and do not provide special comments on the language of the paper. However, the title of this manuscript seems to be not very precise and may I suggest to make some modifies. Transfer learning is a kind of methods which can be applied to different CNN mode and Deep residual networks is a kind of CNN mode. So, it’s strange to have an “and” between “transfer learning”and “deep residual networks”, better be an “of” However, this research conducted many comparative studies among different transfer learning of CNNs, deep residual networks is only a part of it. Also, some machine learning methods had been compared. So, please consider more carefully about the title.

1.2 Abstract

Line 33. I was so surprised by the result that the average accuracy “can reach to 0.85 with only three training sample”. I knew you mean 3 images per each taxon? But it’s better to make it more clearly. AND the accuracy can reach to 0.85 with only 18 training images to retrain the Resnet with not only Fully connected layer but also Conv3_x, Conv4_x and Conv5_x been opened up, it is beyond my expectation. Have you considered overfitting would happen at this situation? It’s only one training result? Or have you randomly picked up training sets several times for training?

1.3 Main text

In the text, I presumed you have only done all kind of training for only one time? Is it repeatable? What would be the result if conducting more times of retraining based on different random training dataset?

Line 206. Can you explain How did F1-score of Resnet come about? In another word, we would like to have more information about how can you directly use Resnet to applied your samples? 

1.4 Figure captions

Line 112-116 Genus name should be in italic.

Author Response

1.1 Title

I am not a native English speaker and do not provide special comments on the language of the paper. However, the title of this manuscript seems to be not very precise and may I suggest to make some modifies. Transfer learning is a kind of methods which can be applied to different CNN mode and Deep residual networks is a kind of CNN mode. So, it’s strange to have an “and” between “transfer learning”and “deep residual networks”, better be an “of”? However, this research conducted many comparative studies among different transfer learning of CNNs, deep residual networks is only a part of it. Also, some machine learning methods had been compared. So, please consider more carefully about the title.

Yes, Deep residual networks is a kind of CNN mode and we test transfer learning on other CNNs and machine learning methods. Final results show that the combination of transfer learning and Deep residual networks had best performance. So we think “and” is OK while “of” may cause misunderstanding. Because the relationship of transfer learning and Deep residual networks is coordinate relationship, transfer learning is not subordinate relationship.

1.2 Abstract

Line 33. I was so surprised by the result that the average accuracy “can reach to 0.85 with only three training sample”. I knew you mean 3 images per each taxon? But it’s better to make it more clearly. AND the accuracy can reach to 0.85 with only 18 training images to retrain the Resnet with not only Fully connected layer but also Conv3_x, Conv4_x and Conv5_x been opened up, it is beyond my expectation. Have you considered overfitting would happen at this situation? It’s only one training result? Or have you randomly picked up training sets several times for training?

Yes, it means 3 images per taxon. We revised the abstract accordingly. Although we chose quite limited number of images per taxon, we used data augmentation to increase the image number so as to avoid overfitting and for each original image, 5 more images can be obtained (Fig.2, line 305-310). We trained each set 10 times using randomly selected images. In our experiments, the opening up of last 4 layers (Conv3_x, Conv4_x and Conv5_x and fully connected layer) is to find an optimized balance between image number and performances. The accuracies of very small training set (1 and 3 images per taxon) did fluctuated more or less and the average results were relatively good.

1.3 Main text

In the text, I presumed you have only done all kind of training for only one time? Is it repeatable? What would be the result if conducting more times of retraining based on different random training dataset?

We performed every training for 10 times with different random images while the test set is the same. The F1-scores are all averaged results.

Line 206. Can you explain How did F1-score of Resnet come about? In another word, we would like to have more information about how can you directly use Resnet to applied your samples? 

F1-score is a popular criterion in AI image recognition with harmonized average of precision and recall (please find it in table 3 in the manuscript). The direct use of Resnet means to use fossil images to tune the parameters of the net itself, while using transfer-learning strategies (TF, TS, TA) means to use fossil images to tune parameters of pre-trained model form ImageNet. The direct use of Resnet shows worse performance and the problem of requiring large training set and was finally excluded.

1.4 Figure captions

Line 112-116 Genus name should be in italic.

       we revised the fonts for italic.

Reviewer 2 Report

Dear authors,

Thank you for allowing me to read a interesting manuscript. I do enjoy AI associated with microfossil identification. Overall the manuscript comprise an interesting technique and aims to tackle a complicated group of microfossils. I would suggested some clarification regarding: 
1) Which rare microfossil are you talking about? Along the main-text is constantly cited soft-bodies, but I do not see any link with the research. I would suggest re-write the objective to something my tangible such as:

- Fast and reliable accurancy for identify microfossils. 

2) You also indicate time consuming on manual selecting microfossils, but, to compare the microfossils you did. What you not did was identifying them manually. Please re-write.

3) on line 187-189, you wrote about the your truly objective.

4) on lines 210-211, you tackled about accuracy. Would sounds better with you add something related to the taphonomical conditions of the samples. You know how the different states of diagenesis can affected the AI identification based on images. 

5) I have add more suggestions in the pdf file. 

You manuscript is really fun to read and will bring light into microfossil identification, therefore, we be better to re-arrange the idea more to a reliable to for ai identification instead of focus on soft-bodies challenge.   

 All the best,

Author Response

1) Which rare microfossil are you talking about? Along the main-text is constantly cited soft-bodies, but I do not see any link with the research. I would suggest re-write the objective to something my tangible such as:

- Fast and reliable accurancy for identify microfossils. 

The “rare microfossils” means some undiscovered types which we still unknown in morphology and taxonomy. Namely, we don’t have such “rare” fossils now. Please find some discovered examples in Han et al 2013, 2017 and Liu et al, 2014 in reference list.

In this paper, we use some common fossils to test the AI capacity to recognized specific fossil type with small training sets. If they can recognize specific fossils accurately, they can help us find more “rare” samples if we will find some relevant fossils and train this AI model with them in the future.

2) You also indicate time consuming on manual selecting microfossils, but, to compare the microfossils you did. What you not did was identifying them manually. Please re-write.

Thanks for mention this point. In our situation, the total taxa of fossils are not too many while the amount of some dominant types could reach more than 90%. So, our major task is to manually selecting different samples form dominant types and do not need identify every sample for specific genus or species. The labour of manually identifying is not the important factor.

3) on line 187-189, you wrote about the your truly objective.

Yes, as we described in point 2) above.

4) on lines 210-211, you tackled about accuracy. Would sounds better with you add something related to the taphonomical conditions of the samples. You know how the different states of diagenesis can affected the AI identification based on images. 

We had a paragraph to introduce relationships between the recognition performance and the biological and taphonomical attributes of fossils. Please see last paragraph of the text (lines 368-377).

In addition, we added some text to state that the fossils we talked about all came from the same locality (Kuanchuanpu Biota), which is a well-known fossil lagerstätten. The preservation quality of most fossils was good as three dimensional exquisite replicas with phosphate and their taphonomical conditions are uniform

5) I have add more suggestions in the pdf file. 

Comments from the PDF file:

Line 2, Which rare microfossils?

please find in point 1) above.

Line 15-16, Are you considering the combination of proxies? Yet, distinct taphonomical conditions, specially specie-specific can hamper the digital identification.

please find in point 4) above.

Line 15, The phrase construction is confusing. Each way of manual selecting are you arguing?

In our situation, the manual selecting means picking up different major types of microfossils by sitting beside a microscope. Please also find in point 2) above.

Line 21, 24, 25, What type of microfossil? carbonaceous? siliceous? It would be good a better detailing.

I am confusing about the concepts. Are you talking about microfossils? or tiny macrofossils? If you are talking about microfossils, are you talking about carbonaceous microfossils? siliceous macrofossils?

All the fossils mentioned here are secondarily phosphatized Cambrian microfossils, generally termed as small shelly fossils, they are quite different from small carbonaceous fossils or siliceous microfossil from deep ocean basin. We deleted “soft body” to be more concise in abstract.

Line 176-179, Despite the high return, you guys manually separated the samples, than took pictures and further ask the ML to recognized the fossils. A awesome improvement, but, still a endless effort. I would suggest you to re-write the phrases related as in line 25, because I improve the velocity of microfossil identification and not the selection under microscope.

The ideal application scenario of the model is that in the future, we may find some good samples by manual selecting at first stage. Then we want to find more of this same type so we seek AI for help. Such good samples would be very rare so we require the AI model have the capacity described in this paper. The effort of manually separating and taking photos for different samples is the simulation of the real application scenario of this AI model.

Line 210, Awesome result. However, still not considering the taphonomical aspects that my affect the images and hamper identification. Would be good if you run the same neurolanguage with distinct taphonomical microfossil states

Please find in point 4) above.

Line 267, How you work is found soft structures? Should erased this line.

Please find published soft body fossil in reference list: Han et al 2013, 2017 and Liu et al, 2014. In order to find more new types of such fossils, we tried to develop this AI model.

Line 273-275, It seems to speculative.

Yes, the molecular clocks data is largely speculative, but with more new fossils discovered in recent years and were incorporated to calibrate the molecular results and the molecular clocks became more accurate in prediction of animal origins. As discussed in Erwin et al 2011, Peterson et al 2009, 2018 in reference list.

Line 286, 384, Agree, but the low potential is no linked to the manual identification. I do not see your AI as a proxy to find the rare taxa, probably can help find fewer among the recovery microfossils.

Please find in explanation of point 1) and line 176-179.

Line 387, I don't think so. You have to prepare the sample, extract and select. At least in this experiment.

Please find in explanation of line 176-179. Namely, in the future, if we find some samples with high research value, we only need very few samples to train the AI and hope it can help us find more. The labour in this experiment is just to tests from multiple aspects.
